# (Generalized) Maximum Cumulative Direct, Residual, and Paired Φ Entropy Approach

**DOI:** 10.3390/e22010091

**Published:** 2020-01-12

**Authors:** Ingo Klein, Monika Doll

**Affiliations:** 1Department of Statistics and Econometrics, Friedrich-Alexander-Universität Erlangen-Nürnberg, Lange Gasse 20, 90403 Nürnberg, Germany; 2GfK SE, Nordwestring 101, 90419 Nürnberg, Germany; monika.doll@fau.de

**Keywords:** cumulative entropy, maximum entropy distribution, generalized Tukey *λ* distribution, generalized logistic distribution, skewed logistic distribution, skewed Tukey *λ* distribution, skewed normal distribution, Weibull distribution, extended Burr XII distribution

## Abstract

A distribution that maximizes an entropy can be found by applying two different principles. On the one hand, Jaynes (1957a,b) formulated the maximum entropy principle (MaxEnt) as the search for a distribution maximizing a given entropy under some given constraints. On the other hand, Kapur (1994) and Kesavan and Kapur (1989) introduced the generalized maximum entropy principle (GMaxEnt) as the derivation of an entropy for which a given distribution has the maximum entropy property under some given constraints. In this paper, both principles were considered for cumulative entropies. Such entropies depend either on the distribution function (direct), on the survival function (residual) or on both (paired). We incorporate cumulative direct, residual, and paired entropies in one approach called cumulative Φ entropies. Maximizing this entropy without any constraints produces an extremely *U*-shaped (=bipolar) distribution. Maximizing the cumulative entropy under the constraints of fixed mean and variance tries to transform a distribution in the direction of a bipolar distribution, as far as it is allowed by the constraints. A bipolar distribution represents so-called contradictory information, which is in contrast to minimum or no information. In the literature, to date, only a few maximum entropy distributions for cumulative entropies have been derived. In this paper, we extended the results to well known flexible distributions (like the generalized logistic distribution) and derived some special distributions (like the skewed logistic, the skewed Tukey λ and the extended Burr XII distribution). The generalized maximum entropy principle was applied to the generalized Tukey λ distribution and the Fechner family of skewed distributions. Finally, cumulative entropies were estimated such that the data was drawn from a maximum entropy distribution. This estimator will be applied to the daily S&P500 returns and time durations between mine explosions.

## 1. Introduction

For a continuous random variable with density *f*, the classical differential (Shannon) entropy is defined by
(1)$$E_{S}\left(f\right)=-\int f\left(x\right)\ln f\left(x\right)dx.$$

Maximizing (Equation 1) with respect to *f* under the constraint of observed power or *L*-moments gives maximum entropy (ME) densities (see e.g., [1,2]). This ME solution represents a distributional model which is compatible with the minimum information given by the fixed constraints. The task of deriving ME densities is important, as they are the only reasonable distribution to use for estimation, as lower entropy distributions would mean to assume information that we do not possess. However, (Equation 1) has some shortcomings in order to be a good measure of information, as it could, for example be negative for special densities [3]. Nevertheless, much of the literature has been concerned with the ME task, not only for classical differential entropy, but also for the cumulative residual, the cumulative, cumulative paired entropy, and further entropies with modifications of the generating functions. Thus, in the following section, a short literature review is given.

Substituting the density by the survival function in (Equation 1) leads to the cumulative residual (Shannon) entropy. Rao et al. [4] were the first who discussed this new entropy. The discussion occurred in the context of reliability theory, where only random variables with non-negative support are of importance while the survival function is the natural distributional concept. The solution of the maximum entropy task under power moment constraints has already been discussed by [4,5]. They used the log-sum inequality to derive the ME solution instead of the usual approach based on the Lagrange–Euler equations. The exponential and more generally, the Weibull distribution, are solutions to these special ME tasks. In the following years, several authors [6,7,8,9,10,11,12,13,14,15,16,17] also focused on cumulative residual entropies. In particular, Drissi et al. [10] were concerned with the ME problem. They considered random variables with support R and derived the logistic distribution as the ME solution under the additional constraint that the ME solution has to be symmetric. DiCrescenzo [18] applied (Equation 1) to the distribution function and called the result ‘cumulative entropy’. Based on early results of DeLuca and Pal [19,20] in the fuzzy set theory concerning membership functions, Li et al. [21] defined a further entropy concept for so-called uncertainty variables which are similar but not identical to random variables. The main idea here is to consider, in (Equation 1), the distribution function as well as the survival function. The obvious corresponding ME task was discussed by [22,23,24]. There are many studies in the literature concerned with the generalization of (Equation 1). Thus, some authors [25,26,27] modified the entropy generating function. General generating functions were considered by [28,29,30] for the definition of the related concept of *f*-divergence [31]. Sometimes *f*-divergences are also called ϕ-divergences [32]. Zografos [33] generalized the cumulative residual Shannon entropy in a similar way. Klein et al. [34] combined the ME task known from uncertainty theory with the use of general entropy generating functions. They derived the Tukey λ distribution as an ME distribution when the entropy generating function of [25] is applied together with the distribution and the survival function. They introduced the term ‘cumulative paired entropy’ analogue to the paired ϕ entropy introduced by [35]. Recent publications [36,37] applied the Havrda-Charvát approach to the survival function under the name ‘Cumulative Tsallis entropy’ of order α. This term refers to the famous paper of [38], where he gives a physical foundation for the Havrda-Charvát approach.

In this paper, the first research question was to clarify what kind of information cumulative entropies really measure. Therefore, we introduce the concept of ‘contradictory information’ in contrast to ‘no information’. As a second research question, we want to unify the diverse approaches of cumulative entropies and their maximization. For this purpose, general cumulative Φ entropies will be introduced. All known variants of cumulative entropies are special cases of this general class. Then, after deriving two general formulas for ME quantile functions under some moment restrictions, we apply these formulas to derive ME distributions for new cumulative entropies (like the cumulative Mielke(r) entropy) as well as to identify the cumulative entropy for some flexible families of distributions that allow for skewness (like the generalized Tukey λ or the generalized logistic distribution). As a byproduct, we find some new families of distributions (like a special skewed Tukey λ distribution and a generalized Weibull distribution). The results are summarized in a table and discussed in detail in the Appendix A. The last research question starts with observed data and tries to estimate the cumulative entropy in such a way that the data come from the corresponding ME distribution. This gives an alternative to non-parametric estimation of density functions or distribution functions.

This paper is organized in line with these research questions. Section 2 starts with the discussion of contradictory information and cumulative entropies in principle. In Section 3, we introduce general cumulative Φ entropies and prove general results for ME distributions for cumulative Φ entropies under different constraints. In Section 4, we propose an estimator for the ME generating function. Finally, we apply this estimator to real datasets. In the Appendix A, we apply the theoretical results to seven families of cumulative Φ entropies (MaxEnt) or families of distributions (GMaxEnt).

## 2. What does Maximizing Cumulative Direct, Residual, and Paired Shannon Entropies Mean?

In this section, we first discuss the concept of ‘contradictory information’ in contrast to no or minimum information and determine that contradictory information corresponds with *U*-shaped/bipolar distributions. Then, we learn that maximizing cumulative paired entropies best reflects this situation by comparing the results with those of maximizing differential, cumulative residual, and cumulative direct entropies. Next, we see that the cumulative residual and cumulative direct entropies do not correspond to a *U*-shaped distribution if the support of a random variable is only non-negative. Overall, in this section, the focus is on Shannon entropies. However, all insights can be transferred to arbitrary cumulative entropies immediately.

The traditional ME approach starts with the result that the uniform distribution has minimum information (= maximum entropy) under the constraint that the area under the density sums up to one. However, there is another concept of maximum entropy in fuzzy set [19] and uncertainty theory [24,39]. Transferring this concept to probability theory, maximum uncertainty represents the fact that an event *A* with probability 0<P(A)<1 and its complementary event $\bar{A}$ with probability $P\left(\bar{A}\right)=1-P\left(A\right)$ have identical probability. This means that P(A)=1/2. Since the Shannon entropy
−P(A)logP(A)−(1−P(A))log(1−P(A))
is maximized for P(A)=1/2, this kind of entropy could serve as the basis for an uncertainty measure. For a continuous random variable *X*, the ensemble of events (X≤x), x∈R such that 0<P(X≤x)<1 can be considered. It is obvious to measure the amount of uncertainty of *X* by
−∫P(X≤x)logP(X≤x)dx−∫(1−P(X≤x))log(1−P(X≤x))dx,
with integration area R. We set 0log0=0. Let *F* be the cumulative distribution function of *X*, then the cumulative paired Shannon entropy is defined by
(2)$$\begin{array}{ccc} CPE_{S}\left(F\right) & = & -\int F(x)\log F(x)dx-\int (1-F(x))log(1-F(x))dx\\ & = & -\int_{0}^{1}\left(u\log u+\left(1-u\right)\log\left(1-u\right)\right)q\left(u\right)du \end{array}$$
with probability integral transformation u=F(x), quantile function $Q\left(u\right)=F^{-1}\left(u\right)$, and quantile density q(u)=dQ(u)/du=1/f(Q(u)) for u∈[0,1]. *f* denotes the density of *X*. If *X* has a compact support [a,b], $CPE_{S}\left(F\right)$ attains its maximum for F(x)=1/2 for a≤x<b. This corresponds to a so-called bipolar distribution with P(X=a)=P(X=b)=1/2. For this bipolar distribution, $CPE_{S}\left(F\right)=\ln2\left(b-a\right)$, a<b holds. Therefore, the cumulative paired Shannon entropy increases with b−a. In contrast to this, the classical differential Shannon entropy ($E_{S}$) takes a value of ln2 for all bipolar distributions, independently of how large the distance between the two mass points is. Rao [5] identified this property as an important advantage of cumulative entropies over the differential entropy. The different behaviors of differential (Shannon) entropy and cumulative Shannon entropy are illustrated in Example 1.

**Example** **1.**
*We consider the symmetric beta distribution with density*
$$f\left(x;\alpha \right)=\frac{1}{B(\alpha ,\alpha )}x^{\alpha -1}{\left(1-x\right)}^{\alpha -1},\;\;0<x<1,\;\alpha >0$$
*while parameter*
α∈(0.1,2)
*. This range allows almost bipolar distributions (*
α=0.1
*), uniform distributions (*
α=1
*), and bell-shaped distributions (*
α=2
*). Figure 1 compares the values of the differential entropy and the cumulative paired Shannon entropy for this range of the parameter α. We see that the differential entropy is non-positive everywhere and attains its maximum for the uniform distribution (*
α=1
*). In contrast to this, the cumulative paired Shannon entropy starts with the maximum value for a bipolar distribution and decreases monotonically with an increase of the parameter α.*


As we do not want to assume information that we do not possess, we perform the ME task to search for densities that are relying on maximum entropy. The densities that are based on minimum information are the only ones that could reasonably be used. In this paper, we propose to rely on bipolar distributions, as they provide contradictory information which is even less useful than minimum or no information for prediction.

The following examples intend to explain where bipolar distributions appear in real situations and how this bipolarity affects the predictability of a random variable *X*.

**Example** **2.**
*In an opinion poll survey, individuals are asked to judge their political belief on a continuous left–right scale. *0* (*100*) symbolizes an extremely left (right) political view. The survey’s result maximizes the cumulative paired Shannon entropy if half of the people state to be extremely left*
(=0)
*, and the other half state to be extremely right*
(=100)
*. This is a situation of maximum uncertainty regarding how to predict the political view of an individual person.*


**Example** **3.**
*If the task is to judge a product on a Likert scale with five ordered categories, the uniform distribution means that no category will be favored by the majority of the voters. However, there could be a result of the voting that is still more confusing than the uniform distribution. What can we learn from the extreme situation that half of the voters give their vote to the best and the other half to the worst category? What does this mean for a new customer thinking over buying the product? In this situation, buying would therefore mean receiving an either excellent or very bad product. This is a situation in which it is most complicated to predict the customer’s decision.*


Both situations of Examples 2 and 3 can be characterized by the term ‘contradictory information’ in contrast to minimum or no information. In general, information is able to reduce uncertainty. However, contradictory information is implicitly defined by the fact that it increases uncertainty and provides a high chance for a wrong decision. (Anti-information is a related, but less formal concept introduced by the information scientist J. Verhoeff [40].) Therefore, as bipolar distributions lead to contradictory information, it is an important task to consider entropies that will be maximized by a bipolar distribution if there are no constraints. Thus, we propose to use cumulative paired entropies to cover contradictory information. Example 1 already showed that the differential entropy does not embody contradictory information. In the following section, we compare the information provided by using cumulative residual and cumulative direct entropies in contrast to using cumulative paired entropies. Rao et al. [4] introduced cumulative residual entropies as a Shannon entropy, where the density is substituted by the survival function. Then, [5,6,7,8,9,10,11,12,13,14,15,16,17] discussed this cumulative residual Shannon entropy:(3)$$CRE_{S}\left(F\right)=-\int_{-\infty }^{\infty }\left(1-F\left(x\right)\right)\ln\left(1-F\left(x\right)\right)dx.$$
DiCrescenzo and Longobardi [41] applied the Shannon entropy to a distribution function and called it cumulative entropy. This gave the formula
(4)$$CDE_{S}\left(F\right)=-\int_{-\infty }^{\infty }F\left(x\right)\ln F\left(x\right)dx.$$
We will call (Equation 4) cumulative direct Shannon entropy ($CDE_{S}\left(F\right)$) for a better distinction to the cumulative residual Shannon entropy ($CRE_{S}\left(F\right)$) and the cumulative paired Shannon entropy ($CPE_{S}\left(F\right)$).

What does maximum entropy mean for the cases of cumulative residual and cumulative direct Shannon entropy? The entropy generating function −(1−u)ln(1−u) attains its maximum for u=1−1/e=0.632>0.5. If the support is [a,b], the maximum $CRE_{S}$ distribution is bipolar. However, this bipolarity is less extreme than in the symmetric case. This is due to that fact that it holds P(X=a)=1−1/e and P(X=b)=1/e. Therefore, there is a preference for the alternative *a* that makes the prediction of *X* easier than in the symmetric case. However, there is still somewhat contradictory information rather than information. Regarding (Equation 4), the probabilities for *a* and *b* have to be interchanged to get a maximum $CDE_{S}$ distribution. The following example illustrates this for a beta distribution with parameters α and β.

**Example** **4.**
*Let X be beta distributed with density*
$$f\left(x;\alpha ,\beta \right)=\frac{1}{B(\alpha ,\beta )}x^{\alpha -1}{\left(1-x\right)}^{\beta -1},\;\;0<x<1,\;\alpha ,\beta >0.$$
*In the following section, we fix β and compute α such that (Equation 3) or (Equation 4) will be maximized. In Table 1*
$\alpha_{cre}$
*and*
$\alpha_{cde}$
*denote the corresponding maximum values. Moreover, this table also contains the maximum values of*
$CRE_{S}$
*and*
$CDE_{S}$
*. We see that the maximum is attained for small values of α and β, denoting a slightly asymmetric U-shaped beta distribution.*

*Figure 2 illustrates the maximum*
$CRE_{S}$
*and*
$CDE_{S}$
*beta distributions for the parameter settings displayed in Table 1.*


To date, the support has been R. However, as, e.g., in the reliability theory, the focus is on random variables with only non-negative support. Thus, it is of importance to also discuss this situation. When only considering a random variable with non-negative support and for the ME quantile function *Q* holds Q(0)=0, maximizing $CRE_{S}$ or $CDE_{S}$ gives a distribution which is no longer *U*-shaped, and the maximum entropy situation no longer corresponds with contradictory information. We illustrate this in Example 5 using a special beta distribution. The parameter β will be set to 1 such that Q(0)=0.

**Example** **5.**
*Let X be beta distributed with density*
$$f\left(x;\alpha \right)=\frac{1}{\alpha }x^{\alpha -1},\;\;0<x<1,\alpha >0.$$
*Table 2 displays the values for*
$CRE_{S}$
*and*
$CDE_{S}$
*for certain values of α. For*
α=0.01
*, we get an extremely right skewed and for*
α=3
*an extremely left skewed distribution. For*
α=0.48
*(*
α=1
*),*
$CRE_{S}$
*(*
$CDE_{S}$
*) attains its maximum.*

*Figure 3 displays, on the top row, the maximum*
$CRE_{S}$
*distribution at*
α=0.48
*and shows that this is an arrangement between an extremely right skewed (*
α=0.01
*) and an extremely left skewed (*
α=3
*) distribution. On the bottom row, we see the maximum*
$CDE_{S}$
*distribution at*
α=1
*.*


The question we raised in the section title on what maximizing cumulative direct, residual, and paired Shannon entropies means can be answered by the conclusion that maximizing these entropies leads to a more or less skewed *U*-shaped distribution as long as there are no special constraints (like Q(0)=0) which are able to prevent this. This *U*-shaped distribution corresponds to contradictory information. Examples 2 and 3 showed that this kind of information is even less useful for prediction and estimation than minimum or no information. Therefore, those distributions are the only reasonable distributions to consider to not assume information that we do not possess.

In the following section, we unify the diverse approaches of cumulative entropies and introduce the general class of cumulative Φ entropies. Then, we derive two general formulas for ME quantile functions under some restrictions.

## 3. Maximum Cumulative Φ Entropy Distributions

In the following, we will first introduce the general class of cumulative Φ entropies that incorporates and generalizes well-known entropies. Then, we will derive general formulas for maximum entropy distributions for this new class regarding arbitrary support as well as non-negative support.

### 3.1. General Class of Cumulative Φ Entropies

In this section section, we incorporate cumulative direct, residual, and paired entropies into one approach. Additionally, instead of focusing on the Shannon case, we allow for a general so-called entropy generating function ϕ, which has to be non-negative and concave on [0,1]. In general, but not mandatory ϕ has a maximum in the interval [0,1]. Hence, the corresponding cumulative ϕ entropies are the cumulative paired ϕ entropy
(5)$$CPE_{\phi }\left(F\right)=\int_{R}\phi \left(F\left(x\right)\right)+\phi \left(1-F\left(x\right)\right)dx=\int_{0}^{1}\left(\phi \left(u\right)+\phi \left(1-u\right)\right))q\left(u\right)du,$$
the cumulative residual ϕ entropy
(6)$$CRE_{\phi }\left(F\right)=\int_{R}\phi \left(1-F\left(x\right)\right)dx=\int_{0}^{1}\phi \left(1-u\right)q\left(u\right)du,$$
and the cumulative direct ϕ entropy with
(7)$$CDE_{\phi }\left(F\right)=\int \phi \left(F\left(x\right)\right)dx=\int_{0}^{1}\phi \left(u\right)q\left(u\right)du.$$

To cover all three cases into one approach, we consider a general concave entropy generating function Φ such that Φ(u)=ϕ(u) or Φ(u)=ϕ(u)+ϕ(1−u) or Φ(u)=ϕ(1−u), u∈[0,1]. Then,
$$CE_{\Phi }\left(F\right)=\int_{0}^{1}\Phi \left(u\right)du$$
will be called cumulative Φ entropy. For the maximum entropy task, the objective is now to maximize this cumulative Φ entropy with respect to *F* under distinct constraints and to search for the distribution that maximizes this entropy. At first, we consider cumulative Φ entropies in a situation with fixed mean and variance. The restriction to these two moments can be explained by the fact that higher moments lead to equations for the ME quantile function which cannot be solved explicitly or the solution does not exist. Then, we discuss the same task with the additional requirement that Q(0)=0. This leads to the fact that the solution can only exist for special relations between the fixed mean and the fixed *k*-th power moment.

In Section 3.2, we consider the situation where mean and variance are fixed and in Section 3.3, the situation with the additional requirement of Q(0)=0.

### 3.2. General Results for Arbitrary Support

The focus in this section is on the situation where mean and variance are fixed and the support is arbitrary. First, the maximum cumulative Φ entropy principle and then the generalized maximum cumulative Φ entropy principle are introduced. General formulas for ME quantile functions are provided.

#### 3.2.1. Maximum Cumulative Φ Entropy Approach

In this section, for a given entropy and fixed constraints, general formulas for ME distributions are derived. This maximum cumulative Φ entropy approach follows the maximum entropy principle in the sense of [42,43]. The following theorem provides a general formula for the ME quantile function *Q*.

**Theorem** **1.**
*Let*
$CE_{\Phi }$
*be the cumulative *Φ* entropy with concave entropy generating function *Φ* such that the derivative*
$\Phi^{\prime }$
*exists i.e., is quadratic integrable over*
[0,1]
*, and*
|Φ(0)|<∞
*,*
|Φ(1)|<∞
*hold. Then, the maximum*
CE
*distribution under the constraints of fixed mean μ and variance*
$\sigma^{2}$
*is given by the quantile function*
(8)$$Q\left(u\right)=\mu +\sigma \frac{-\Phi^{\prime }\left(u\right)+\left(\Phi \left(1\right)-\Phi \left(0\right)\right)}{\sqrt{\int_{0}^{1}\Phi^{\prime }{\left(u\right)}^{2}du+{\left(\Phi \left(1\right)-\Phi \left(0\right)\right)}^{2}}}.$$


**Proof.** The objective function
$$\int \Phi \left(F\left(x\right)\right)du=\int_{0}^{1}\Phi \left(u\right)q\left(u\right)du$$
has to be maximized under the restrictions of fixed
$$\mu =\int_{0}^{1}Q\left(u\right)du\;\;and\;\;\mu^{2}+\sigma^{2}=\int_{0}^{1}Q{\left(u\right)}^{2}du$$
with respect to the quantile function *Q* and the quantile density *q*. This leads to the Lagrange function
$$\begin{array}{ccc} L(q,Q,\lambda_{1},\lambda_{2}) & = & \int_{0}^{1}\Phi \left(u\right)q\left(u\right)du-l_{1}\left(\int_{0}^{1}Q\left(u\right)du-\mu \right)\\ & & -l_{2}\left(\int_{0}^{1}Q{\left(u\right)}^{2}du-\sigma^{2}+\mu^{2}\right), \end{array}$$
with $l_{1}$ and $l_{2}$ denoting the Lagrange parameters. The Euler–Lagrange equation gives
$$\frac{d}{du}\frac{\partial L}{\partial q}-\frac{\partial L}{\partial Q}=\Phi^{\prime }\left(u\right)+l_{1}+2l_{2}Q\left(u\right)\overset{!}{=}0.$$
Solving this equation leads to the quantile function
$$Q\left(u\right)=\frac{1}{2l_{2}}\left(-\Phi^{\prime }\left(u\right)-l_{1}\right).$$
$l_{1}$ and $l_{2}$ are determined by the moments μ and $\sigma^{2}$. Rearranging
$$\mu =\frac{1}{2l_{2}}\int_{0}^{1}\left(-\Phi^{\prime }\left(u\right)-l_{1}\right)du=-\frac{1}{2l_{2}}\left(\Phi (1)-\Phi (0)\right)-\frac{l_{1}}{2l_{2}}$$
leads to
$$l_{1}=-2l_{2}\mu -\left(\Phi \left(1\right)-\Phi \left(0\right)\right)$$
and
(9)$$Q\left(u\right)=\mu -\frac{1}{2l_{2}}\left(\Phi^{\prime }\left(u\right)-\left(\Phi \left(1\right)-\Phi \left(0\right)\right)\right).$$
From
$$\begin{array}{ccc} \mu^{2}+\sigma^{2} & = & \int_{0}^{1}Q{\left(u\right)}^{2}\;du\\ & = & \mu^{2}+\frac{1}{l_{2}}\mu \left(\int_{0}^{1}\Phi^{\prime }\left(u\right)-\left(\Phi \left(1\right)-\Phi \left(0\right)\right)du\right)\\ & & +\frac{1}{4l_{2}^{2}}\int_{0}^{1}{\left(\Phi^{\prime }\left(u\right)-\left(\Phi \left(1\right)-\Phi \left(0\right)\right)\right)}^{2}\;du\\ & = & \mu^{2}+\frac{1}{4l_{2}^{2}}\left(\int_{0}^{1}\Phi^{\prime }{\left(u\right)}^{2}du-2\int_{0}^{1}\Phi^{\prime }\left(u\right)du\left(\Phi \left(1\right)-\Phi \left(0\right)\right)+{\left(\Phi \left(1\right)-\Phi \left(0\right)\right)}^{2}\right)\\ & = & \mu^{2}+\frac{1}{4l_{2}^{2}}\left(\int_{0}^{1}\Phi^{\prime }{\left(u\right)}^{2}du-{\left(\Phi \left(1\right)-\Phi \left(0\right)\right)}^{2}\right). \end{array}$$
Solving with respect to $l_{2}$ leads to
$$l_{2}=\frac{1}{2\sigma }\sqrt{\int_{0}^{1}\Phi^{\prime }{\left(u\right)}^{2}du-{\left(\Phi \left(1\right)-\Phi \left(0\right)\right)}^{2}}.$$
Inserting $l_{2}$ into (Equation 9) gives the quantile function (Equation 8). □

#### 3.2.2. Generalized Maximum Cumulative Φ Entropy Approach

In this section, for a given quantile function *Q*, the corresponding generating function Φ of the cumulative Φ entropy will be derived. This generalized maximum cumulative Φ entropy approach follows the generalized maximum entropy principle formulated by [44]. We also use formula (Equation 8) for this approach. For a simpler notation, we introduce the partial mean function. Let *X* be the random variable corresponding to *Q* and *f* be the density of *X*. Thus, the partial mean function μ(u) is given by
(10)$$\begin{array}{ccc} \mu (u) & = & \int_{0}^{u}Q\left(v\right)dv=\int_{-\infty }^{Q(u)}xf\left(x\right)dx=-E\left(X|X\leq Q\left(u\right)\right)P\left(X\leq Q\left(u\right)\right),\\ & & =uE(X|X\leq Q(u))\;\;u\in [0,1]. \end{array}$$
Obviously, μ(0)=0 and μ(1)=μ hold.

The following corollary states that the negative of the partial mean function determines the entropy generating function such hat *Q* is the ME quantile function under the constraints of given mean μ and variance $\sigma^{2}$.

**Corollary** **1.**
*Let Q be a quantile function. The entropy generating function *Φ*, such that Q is ME under the constraints of given mean and variance, is given by*
Φ(u)=−μ(u)
*,*
u∈[0,1]
*.*


**Proof.** Setting $\Phi^{\prime }\left(u\right)=-Q\left(u\right)$, u∈[0,1] gives
(11)$$\Phi \left(u\right)=-\int_{0}^{u}Q\left(v\right)dv=-\mu \left(u\right)\;\;u\in \left[0,1\right].$$
 □

Hence, −Φ(u)/u is the conditional mean of *X* given X≤Q(u) for u∈[0,1]. It holds μ(0)=0 and μ(1)=μ such that Φ(0)=0 and Φ(1)=−μ. The partial mean function μ(u) therefore has a special role. As μ(u) sums up the values *x* of *X* weighted with the density f(x) until the *u*-quantile of *X*, this addition gives constant values until the median quantile for an extremely *U*-shaped distribution. Thereafter, the value will be changed one time and stays again constant. Thus, the heavier the tails of a distribution, the steeper the entropy-generating function Φ(u) at u=0 and u=1. This leads to a large value for the derivative $\Phi^{\prime }\left(u\right)$ at u=0 and u=1. If the support is R, then $\lim\Phi^{\prime }\left(0\right)=\infty$ and $\lim\Phi^{\prime }\left(u\right)=-\infty$. In line with the generalized maximum entropy principle, we will use (Equation 11) to derive Φ such that a given distribution has the ME property under the constraints of fixed mean and variance. In Section 5, based on (Equation 11), we will propose an estimator for Φ.

### 3.3. General Results for Non-Negative Support

To date, in the literature, the ME task was mainly considered for lifetime distributions with the special property that the support is (0,∞). Therefore, in this section, the focus is on the situation where next to the constraints of fixed mean and variance also the support is restricted to (0,∞). Similar to Section 3.2, the maximum cumulative Φ entropy principle and then the generalized maximum cumulative Φ entropy principle will be introduced in this situation and general formulas for ME quantile functions will be provided.

#### 3.3.1. Maximum Cumulative Φ Entropy Approach

In this section, for given entropy and constraints, a general formula for ME distributions will be derived, following the maximum cumulative Φ entropy approach, while the support of the ME distribution is (0,∞), which means that Q(0)=0 holds for the ME quantile function *Q*. From this fact, we get an additional constraint for the ME task. As further constraints, we consider a fixed mean μ and a fixed *k*-th power moment $\mu_{k}^{\prime }$, k>1. The following theorem shows how to derive the ME quantile function under these three constraints. For an ME solution to be existent, it requires a special relationship between the fixed moments μ and $\mu_{k}^{\prime }$.

**Theorem** **2.**
*Let *Φ* a be concave function on*
[0,1]
*with derivative*
$\Phi^{\prime }$
*such that*
$-\Phi^{\prime }\left(u\right)+\Phi^{\prime }\left(0\right)$
*is monotonically increasing. Then, the ME quantile function under the constraints of given mean and k-th power moment*
$\mu_{k}^{\prime }$
*is*
(12)$$Q\left(u\right)=\frac{{\left(-\Phi^{\prime }\left(u\right)+\Phi^{\prime }\left(0\right)\right)}^{1/(k-1)}}{\int_{0}^{1}{\left(-\Phi^{\prime }\left(u\right)+\Phi^{\prime }\left(0\right)\right)}^{1/(k-1)}du}\mu ,$$
*if*
(13)$$\frac{\mu_{k}^{\prime }}{\mu^{k}}=\frac{\int_{0}^{1}{\left(-\Phi^{\prime }\left(u\right)+\Phi^{\prime }\left(0\right)\right)}^{k/(k-1)}du}{{\left(\int_{0}^{1}{\left(-\Phi^{\prime }\left(u\right)+\Phi^{\prime }\left(0\right)\right)}^{1/(k-1)}du\right)}^{k}}.$$
*Otherwise there is no solution of the ME task.*


**Proof.** Due to the Euler–Lagrange equation, it is
$$\Phi^{\prime }\left(u\right)+l_{1}+kl_{2}Q{\left(u\right)}^{k-1}\overset{!}{=}0,\;\;u\in \left[0,1\right].$$
The constraint Q(0)=0 leads to $l_{1}=-\Phi^{\prime }\left(0\right)$ and
$$Q\left(u\right)={\left(\left(-\Phi^{\prime }\left(u\right)+\Phi^{\prime }\left(0\right)\right)/\left(kl_{2}\right)\right)}^{1/(k-1)},\;\;u\in \left[0,1\right].$$
$kl_{2}$ can be derived from
$$\mu =\int_{0}^{1}Q\left(u\right)du={\left(\frac{1}{kl_{2}}\right)}^{1/(k-1)}\int_{0}^{1}{\left(-\Phi^{\prime }\left(u\right)+\Phi^{\prime }\left(0\right)\right)}^{1/(k-1)}du$$
as
$$kl_{2}={\left(\frac{1}{\mu }\right)}^{k-1}{\left(\int_{0}^{1}{\left(-\Phi^{\prime }\left(u\right)+\Phi^{\prime }\left(0\right)\right)}^{1/(k-1)}du\right)}^{k-1}.$$
Inserting $kl_{2}$ into Q(u) gives (Equation 12) immediately.There is the third constraint
$$\mu_{k}^{\prime }=\int_{0}^{1}Q{\left(u\right)}^{k}du=\frac{\int_{0}^{1}{\left(-\Phi^{\prime }\left(u\right)+\Phi^{\prime }\left(0\right)\right)}^{k/(k-1)}du}{{\left(\int_{0}^{1}{\left(-\Phi^{\prime }\left(u\right)+\Phi^{\prime }\left(0\right)\right)}^{1/(k-1)}du\right)}^{k}}\mu^{k}.$$
Dividing $\mu^{k}$ on both sides gives (Equation 13). □

In the most popular case, mean and variance are fixed. This means k=2 and
$$Q\left(u\right)=\frac{-\Phi^{\prime }\left(u\right)+\Phi \left(0\right)}{\int_{0}^{1}\left(-\Phi^{\prime }\left(u\right)+\Phi^{\prime }\left(0\right)\right)du}\mu =\frac{-\Phi^{\prime }\left(u\right)+\Phi^{\prime }\left(0\right)}{-\Phi \left(1\right)+\Phi \left(0\right)+\Phi^{\prime }\left(0\right)}\mu$$
and
$$\begin{array}{ccc} \frac{\mu_{2}^{\prime }}{\mu^{2}} & = & \frac{\int_{0}^{1}{\left(-\Phi^{\prime }\left(u\right)+\Phi^{\prime }\left(0\right)\right)}^{2}du}{{\left(\int_{0}^{1}\left(-\Phi^{\prime }\left(u\right)+\Phi^{\prime }\left(0\right)\right)du\right)}^{2}}\\ & = & \frac{\int_{0}^{1}\Phi^{\prime }{\left(u\right)}^{2}du-2\left(\Phi \left(1\right)-\Phi \left(0\right)\right)\Phi^{\prime }\left(0\right)+\Phi^{\prime }{\left(0\right)}^{2}}{{\left(\Phi \left(1\right)-\Phi \left(0\right)\right)}^{2}-2\left(\Phi \left(1\right)-\Phi \left(0\right)\right)\Phi^{\prime }\left(0\right)+\Phi^{\prime }{\left(0\right)}^{2}} \end{array}$$

#### 3.3.2. Generalized Maximum Cumulative Φ Entropy Approach

The generalized maximum cumulative Φ entropy approach for random variables with non-negative support and Q(0)=0 remains to be discussed. We start with the knowledge of the quantile function *Q* to derive the corresponding generating function Φ of the cumulative Φ entropy such that *Q* is the ME quantile function for Φ under the constraints Q(0)=0 and fixed mean μ and fixed *k*-th power moment $\mu_{k}^{\prime }$. Therefore, we introduce a special partial mean function. $\mu_{k-1}\left(u\right)$ denotes the partial (k−1)-th power mean function with
$$\mu_{k-1}\left(u\right)=uE\left[X^{k-1}|X\leq Q\left(u\right)\right],\;\;u\in \left[0,1\right]$$
for k=2,3,…. This partial (k−1)-th power moment function is an important part of the entropy generating function as the following corollary shows.

**Corollary** **2.**
*Let Q be a quantile function. The entropy generating function *Φ*, such that Q is ME under the constraints*
Q(0)=0
*, fixed mean and fixed variance, is given by*
$$\Phi \left(u\right)=\mu_{k-1}^{\prime }u-\mu_{k-1}\left(u\right),\;\;u\in \left[0,1\right].$$


**Proof.** Let *X* be the random variable corresponding to *Q* and *f* be the density of *X*. From
$$Q\left(u\right)={\left(-\Phi^{\prime }\left(u\right)-\Phi^{\prime }\left(0\right)\right)}^{1/(k-1)},\;\;u\in \left[0,1\right]$$
we get
$$-\Phi \left(u\right)+\Phi^{\prime }\left(0\right)u=\int_{0}^{u}Q{\left(v\right)}^{k-1}dv\equiv \mu_{k-1}\left(u\right),\;\;u\in \left[0,1\right].$$
It is easy to verify that Φ(0)=0. Under the assumption Φ(1)=0 it is $\Phi^{\prime }\left(0\right)=\mu_{k-1}^{\prime }$ and
(14)$$\Phi \left(u\right)=\mu_{k-1}^{\prime }u-\mu_{k-1}\left(u\right),\;\;u\in \left[0,1\right].$$
 □

In Section 5, we use (Equation 14) to estimate Φ from a data set such that the data are generated by the corresponding ME distribution under the constraints of Q(0)=0 and fixed mean and fixed *k*-th power moment.

## 4. Applications

In this section, we give an overview about some ME distributions for cumulative entropies applying the results of Section 3. For some choices of Φ, the problem of the ME task has already been solved. In the following section, we consider further choices of Φ with a focus on those that lead to well known distributions. With the ME principle, it is no problem to generate completely new distributions, but this will not be the objective of this paper.

Table 3 displays an overview of several entropy generating functions and the corresponding ME distributions. The table is divided by the situation where mean and variance are fixed, by the distinction of the MaxEnt and the GMaxEnt task, and by the situation with the additional requirement of Q(0)=0. Moreover, while cases no. 1 to no. 4 require symmetry of the ME distribution, cases no. 5 to no. 12 allow for skewness of the ME distribution. $f_{N}$ and $F_{N}^{-1}$ denote the density and the quantile function of the standard normal distribution. We try to assign well known terms to the cumulative entropies generated by the respective Φ. For the solution of the GMaxEnt task (no. 9 and 10), such terms are not available. The second column refers to the Appendix where the cases are discussed in detail.

Some of the results presented in Table 3 are already known from the literature. These are the solutions of no. 1 [24], no. 2 [34], no. 3 [34], no. 4 [23,45] and no. 11 [5]. The remaining cases state new results which are discussed in the Appendix for all readers interested in flexible statistical distributions. The general finding can best be illustrated by solutions no. 1 and no. 2. The ME distributions are the logistic and the Tukey λ distribution. Solving the ME task for the classical differential and the Havrda-Charvát (or Tsallis) entropy given fixed mean and variance results in the normal and the *t*- or *r*-distribution [46,47]. The difference is easy to explain. The cumulative entropy pulls the ME distribution as much as possible (limited by the restrictions) towards a *U*-shaped distribution. This leads to distributions with heavier tails (logistic instead of normal, Tukey λ instead of *t* or *r*).

There are a lot of entropy-generating functions well-known from physics which could also be considered in the context of cumulative entropies. It is easy to show that the results of Theorem 1 and Theorem 2 can be applied to, e.g., the generating functions of the Rényi [27], the Kaniadakis [48], or the Hanel-Thurner entropy [49], to mention only a few. Another comment deals with the concept of skewness. Some families of distributions have natural parameters of skewness. If the members of these families have closed expressions for the quantile function, Corollary 1 can be applied directly to derive the function Φ for the corresponding cumulative entropy (GMaxEnt task). This is the reason why we focus on the generalized Tukey λ distribution. It is worth noting that again, a kind of non-symmetric cumulative Havrda-Charvát entropy appears as solution (see no 9). Other families of skewed distributions will be defined by modifying a given symmetric distribution. The Fechner approach as well as the still more popular Azzalini approach proceed in such a way. The Fechner approach introduces skewness by splitting the scale parameter for the positive and the negative halves of the underlying symmetric distribution. This leads to a corresponding splitting of the quantile function. Corollary 1 can again easily be applied to solve the GMaxEnt task as long as the quantile function is available in a manageable form. The solution for the normal distribution is given by solution no. 10. For the more popular Azzalini approach [50,51], this is not the case. Therefore, we omit to discuss the GMaxEnt task for this concept to generate skewed distributions.

Table 3 only contains special choices of the entropy generating functions Φ. The main question is how to know Φ. The answer could be given by an axiomatic approach or empirically. The starting point for the axiomatic approach are fundamental requirements with a plausible and general accepted interpretation in the considered scientific discipline. Such axiomatizations are available for the differential and the Tsallis entropy. A recent publication on this topic is e.g., [52]. In the context of cumulative entropies, we can go back to approaches in the fuzzy set theory. In this theory, measures of indefiniteness will be axiomatized (see [19,20,53]). The axioms are directly applicable to cumulative entropies. (The discussion of alternative entropies, skewness and axiomatic approaches is based on valuable comments of two anonymous referees.) In the following section, we do not want to discuss the axiomatic approach further. Instead, in the next section, we will focus on how to estimate the entropy generating function Φ.

## 5. Estimating the Entropy Generating Function

Can we learn something from data about the entropy generating function Φ for which the data generating distribution is an ME distribution under the constraints of given mean and variance? The entropy generating function Φ is given by the partial mean function
μ(u)=uE[X|X≤Q(u)]u∈[0,1].
Therefore, we can estimate this partial mean function to get an estimator for Φ.

Let $X_{1},\ldots ,X_{n}$ be identically and stochastically independent distributed random variables. $X_{\left(n:1\right)},\ldots ,X_{\left(n:n\right)}$ denote the corresponding sequence of order statistics. For a fixed value u∈[0,1] such that nu∈{1,2,…,n} we consider an estimator of the form
$$\hat{\Phi }\left(u\right)=-\hat{\mu }\left(u\right)=-u\frac{1}{nu}\sum_{i=1}^{nu}X_{\left(n:i\right)},\;\;nu\in \left\{1,2,\ldots ,n\right\}$$
for the entropy generating function Φ.

We demonstrate the usefulness of this estimator by the following examples.

**Example** **6.**
*The data set consists of the S&P500 standardized daily logarithmic returns from 10-05-2012 to 10-04-2017. (The data are available from https://fred.stlouisfed.org/series/SP500.) This gives 1256 data points. We have to notice that the mean and the variance are fixed to the values *0* and *1*. In Figure 4, we compare the estimated entropy generating function (*
$-\hat{\mu }\left(u\right)$
*) with the entropy generating functions of the standardized t distribution with *4* degrees of freedom and the standard normal distribution. Standardizing gives also the mean value *0* and the variance *1* for the t distribution. The entropy generating function of the t distribution must be calculated by numerical integration. We chose the number of degrees of freedom by trail and error, but ML estimation gives a value not far away from *4*.*

*We can see that by estimating the entropy generating function *Φ* by the partial mean function, the density of the S&P500 standardized daily logarithmic returns can be fitted quite well.*


In the following example, we consider a situation with non-negative support. We know from (Equation 14) that for a non-negative random variable with Q(0)=0 and fixed mean μ and fixed *k*-th power moment $\mu_{k}^{\prime }$ the entropy generating function Φ is given by
$$\Phi \left(u\right)=\mu_{k-1}^{\prime }u-\mu_{k-1}\left(u\right)=\mu_{k-1}^{\prime }u-uE\left[X^{k-1}|X\leq Q\left(u\right)\right].$$
For this entropy generating function *Q* is an ME quantile function.

To get an estimator for Φ, it is only necessary to estimate the (k−1)-th power mean $\mu_{k-1}^{\prime }$ and the partial (k−1)-th power mean function $\mu_{k-1}\left(u\right)$. For a fixed value u∈[0,1] (such that nu∈{1,2,…,n}), a natural estimator for the partial (k−1)-th power mean function is
$${\hat{\mu }}_{k-1}\left(u\right)=u\frac{1}{nu}\sum_{i=1}^{nu}X_{\left(n:i\right)}^{k-1},\;\;nu\in \left\{1,2,\ldots ,n\right\}.$$
An estimator for the entropy generating function Φ is given by
$$\hat{\Phi }\left(u\right)=u\left(\frac{1}{n}\sum_{i=1}^{n}X_{\left(n:i\right)}^{k-1}-\frac{1}{nu}\sum_{i=1}^{nu}X_{\left(n:i\right)}^{k-1}\right),\;\;nu\in \left\{1,2,\ldots ,n\right\}.$$

We will show that this estimator works well for a real data set and the Weibull distribution. Therefore, we need the partial (k−1)-th power mean function for the Weibull distribution with shape parameter *r* and scale parameter λ. For this distribution, it holds
$$\mu_{k-1}\left(u\right)=\int_{0}^{u}{\left(\lambda (-{\left(1-v\right)}^{r})\right)}^{k-1}dv=\lambda^{k-1}\Gamma \left(\frac{k-1}{r}+1\right)\Gamma \left(-\ln\left(1-v\right);\frac{k-1}{r}+1,1\right)$$
for u∈[0,1]. Γ(x;a,b) denotes the distribution function of a Γ distribution with shape parameter *a* and scale parameter β. The corresponding entropy generating function Φ such that this Weibull distribution is $CE_{\Phi }$ maximum under Q(0)=0 and the constraints of fixed mean μ and fixed *k*-the power moment is
$$\Phi \left(u\right)=u\lambda^{k-1}\Gamma \left(1+\frac{k-1}{r}\right)\left(1-\Gamma \left(-\ln\left(1-u\right);\frac{k-1}{r}+1,1\right)\right),\;\;u\in \left[0,1\right].$$
*k* determines the shape parameter *r* by the relation
$$\frac{\mu_{k}^{\prime }}{\mu^{k}}=\frac{\Gamma (1+k/r)}{\Gamma {\left(1+1/r\right)}^{k}}.$$

**Example** **7.**
*Let X be a random variable representing the duration in days between two explosions in the mines of a specific region. From [54], we get the following dataset with the duration between *41* mine explosions:*
$$\begin{array}{cccccccccc} 378 & 36 & 15 & 31 & 215 & 11 & 137 & 4 & 15 & 72\\ 96 & 124 & 50 & 120 & 203 & 176 & 55 & 93 & 59 & 315\\ 59 & 61 & 1 & 13 & 189 & 345 & 20 & 81 & 286 & 114\\ 108 & 188 & 233 & 28 & 22 & 61 & 78 & 99 & 326 & 275 \end{array}$$

*We set*
k=2
*. This means that for every potential ME distribution,*
$\mu_{2}^{\prime }/\mu^{2}=1.762$
*has to hold. This implies*
r=1.148
*for the shape parameter r of the Weibull distribution. In Figure 5, the estimated entropy generating function is compared with*
$\mu_{k-1}\left(u\right)$
*for this Weibull distribution. The fit seems to be rather good in view of the relatively small sample size.*


Further work will be conducted to estimate the number of degrees or parameters of other flexible distributions by minimizing the distance between the easy-to-calculate empirical entropy generating function, and the entropy-generating function of the distribution, we suppose the data could be generated from. The advantage of this procedure could be that the empirical entropy-generating function is rather smooth. Therefore, minimizing the distance between the entropy generating functions could be more accurate than considering the distance between the empirical quantile functions, a density estimator, or the empirical distribution functions and the corresponding theoretical counterpart. However, this will be investigated in future research.

## 6. Conclusions

To be able to estimate and predict while not using information that we do not possess, it is important to derive maximum entropy distributions. Maximizing Shannon’s differential entropy under different moment constraints is a well-known task. Without any constraints, the differential entropy will be maximized by a uniform distribution representing the situation of no information. However, an extremely bimodal (=bipolar) distribution represents a situation of so-called contradictory information since an event and its complement can happen with equal probability. In this situation, it is extremely hard to make a forecast, even harder than for a uniformly distributed random variable. Hence, this paper claims that contradictory information is even less useful than minimum or no information as it increases uncertainty and provides a high chance for a wrong decision. Such a bipolardistribution is covered by maximizing a cumulative entropy instead of the differential entropy without any constraints. Such a cumulative entropy depends either on the distribution function (direct), on the survival function (residual) or on both (paired). Under the constraints of fixed mean and variance, maximizing the cumulative entropy tries to transform a distribution in the direction of a bipolar distribution as far as it is allowed by the constraints. For so-called cumulative paired entropies and the constraints that mean and variance are known, solving the maximization problem leads to symmetric ME distributions like the logistic and the Tukey λ distribution [21,34]. So far, other ME distributions were found for the cumulative paired Leik and Gini entropy [23,34,45]. There are two different principles to derive maximum entropy distributions. The maximum entropy principle in the sense of [42,43] is the task to derive an ME distribution for a given entropy and fixed constraints. The generalized maximum entropy approach formulated by [44] uses a given ME distribution for which the corresponding generating function of the cumulative entropy will be derived. In this paper, we will applied both approaches for the cumulative Φ entropy, which generalizes the cumulative paired entropy in several ways and thus introduced the maximum cumulative Φ entropy approach and the generalized maximum cumulative Φ entropy approach. Moreover, we regarded situations with different constraints. First, we considered a situation with arbitrary support and given mean and variance and second a situation with non-negative support and the additional constraint of Q(0)=0 for the ME quantile function. This was done, as in the literature the ME task was considered mainly for lifetime distributions with the special property that the support is [0,∞) and it holds Q(0)=0. Under these additional constraints, we derived ME distributions for fixed mean and *k*-th power moment. For the situation with arbitrary support and given mean and variance, we introduced the cumulative paired Mielke(r) entropy and derived the ME distributions. The results already known for the cumulative paired Leik and Gini entropy are included for r=1 and r=2. Then, starting with a natural generalization of the derivative of the entropy generating function known from the logistic distribution, we derived as ME distribution the generalized logistic distribution (GLO) immediately. Considering a linear combination of entropy generating functions led to new ME distributions with skewness properties. Here, we derived the skewed logistic distribution and the skewed Tukey λ distribution in line with [55]. Next, using the generalized maximum Φ entropy approach, we derived an entropy generating function such that a pre-specified skewed distribution is an ME distribution. The generalized Tukey λ distribution served as an example. Now, we considered Fechner’s proposal to define different values of a scale parameter for both halves of a distribution for getting skewed distributions. Again, we derived the corresponding entropy generating function. The skewed normal distribution served as an illustrative example. Then, we focused on the situation where the support of the ME distribution is restricted to (0,∞), while using the maximum cumulative Φ entropy approach. Here, we derived as ME distribution for the cumulative residual Shannon entropy the Weibull distribution and for the cumulative residual Havrda-Charvát entropy the extended Burr XII distribution. Finally, we proposed an estimator for the cumulative Φ entropy generating function representing all the properties of the underlying ME data generating distribution. This gives an alternative to non-parametric estimation of density functions or distribution functions. The usefulness of this estimator was demonstrated for two real data sets.

## Figures and Tables

**Figure 1 entropy-22-00091-f001:**
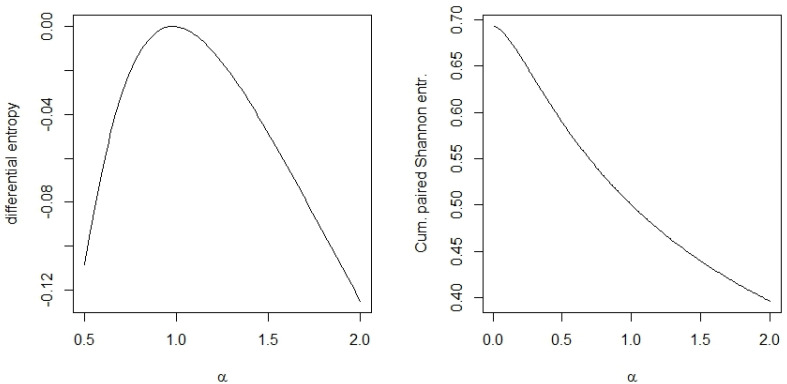
Differential entropy and cumulative paired Shannon entropy for the symmetric beta distribution with several parameter values of *α*.

**Figure 2 entropy-22-00091-f002:**
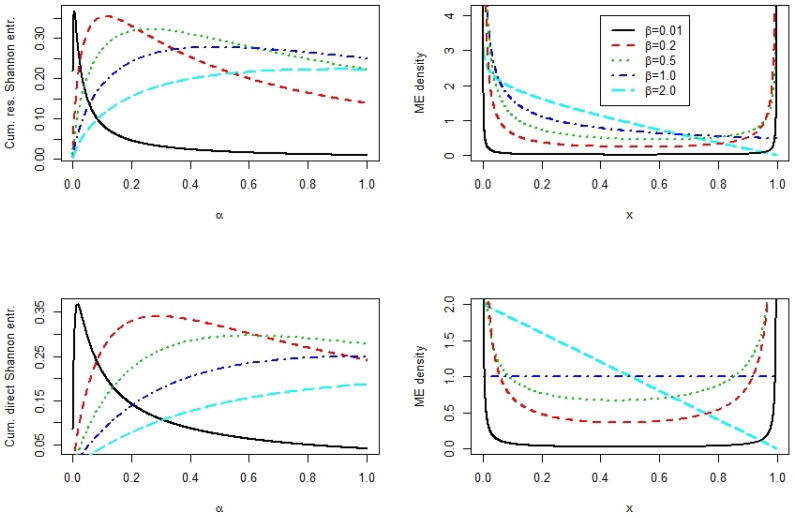
Cumulative residual and cumulative direct Shannon entropy for the asymmetric beta distribution with parameter values *α* such that $CRE_{S}$ ($CDE_{S})$ is maximized for given *β*.

**Figure 3 entropy-22-00091-f003:**
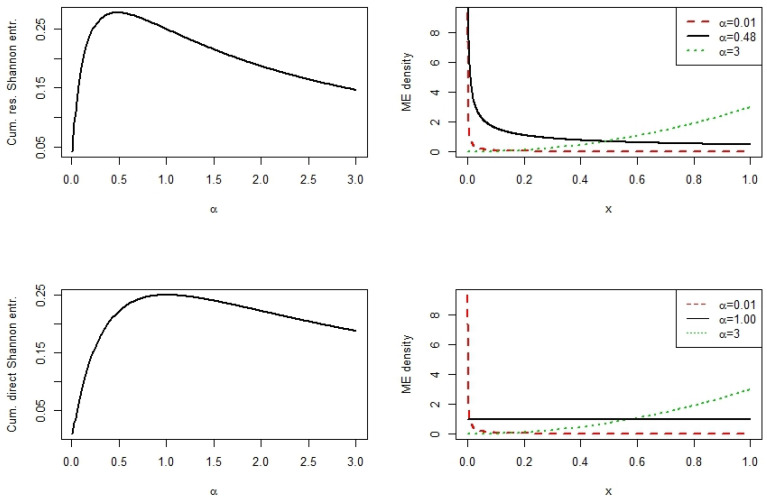
Cumulative residual and cumulative direct Shannon entropy and density for the beta distribution with several parameter values *α* and β=1.

**Figure 4 entropy-22-00091-f004:**
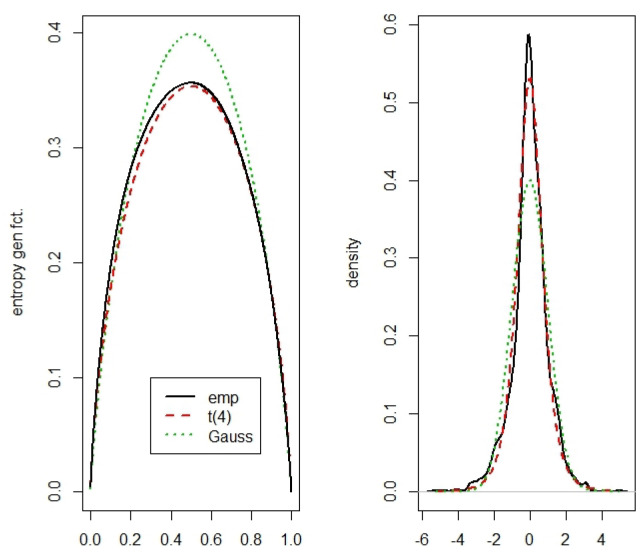
Estimated entropy generating function and estimated density for the S&P500 standardized daily log returns from 10-05-2012 to 10-04-2017.

**Figure 5 entropy-22-00091-f005:**
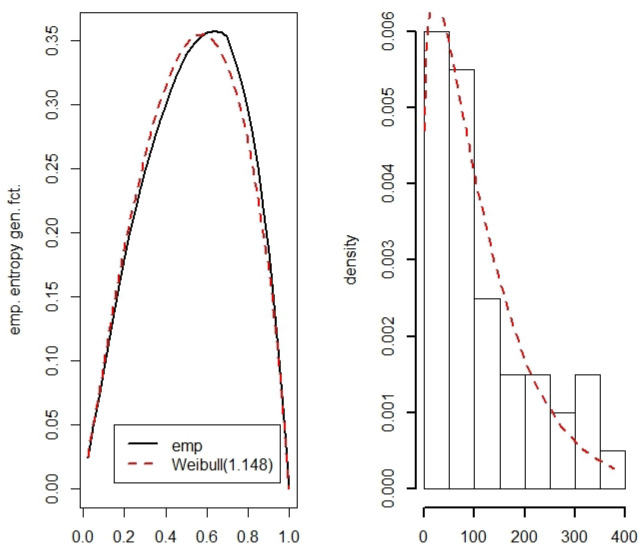
Estimated entropy generating function and estimated density for a data set with time intervals between 41 mine explosions.

**Table 1 entropy-22-00091-t001:** Maximum entropy beta distributions with several parameter values of *β*.

β	$\alpha_{\mathrm{cre}}$	Max. ${\mathrm{CRE}}_{S}$	$\alpha_{\mathrm{cde}}$	Max. ${\mathrm{CDE}}_{S}$
0.01	0.006	0.3678	0.017	0.3677
0.20	0.111	0.3543	0.290	0.3408
0.50	0.259	0.3222	0.595	0.2970
1.00	0.482	0.2778	1.000	0.2500
2.00	0.905	0.2226	1.000	0.1869

**Table 2 entropy-22-00091-t002:** Cumulative residual and cumulative direct Shannon entropy for the beta distribution with several parameter values α and β=1.

α	${\mathrm{CRS}}_{S}$	${\mathrm{CDE}}_{S}$
0.01	0.1836	0.0826
0.48	0.2779	0.2191
1.00	0.2500	0.2500
3.00	0.1464	0.1876

**Table 3 entropy-22-00091-t003:** Entropy generating functions with corresponding maximum entropy distributions.

No.	App.		Φ	ME distr.
Fixed mean and variance, without Q(0)=0				
1		Shannon	−ulnu−(1−u)ln(1−u)	logistic
2		Havrda-Charvát	$u\left(u^{\alpha -1}-1\right)/\left(1-\alpha \right)$	Tukey λ
			$+\left(1-u\right)({\left(1-u\right)}^{\alpha -1}-1)/\left(1-\alpha \right)$	
3	Section A.1	Leik	2(1/2−|u−1/2|)	bimodal
4	Section A.1	Gini	$2(1/4-{\left(u-1/2\right)}^{2}$	uniform
5	Section A.1	Mielke	$2({\left(1/2\right)}^{r}-|u-1/2{|}^{r}$	symm. beta
6	Section A.2	Havrda-Charvát	$-\frac{B(2-\alpha ,\alpha )}{\alpha -1}\left(\beta \left(u;2-\alpha ,\alpha \right)-\frac{u}{B(2-\alpha ,\alpha )}\right)$	general. logistic
		like		
7	Section A.3	non-symm.	$-\alpha_{1}u\ln u-\alpha_{2}\left(1-u\right)\ln\left(1-u\right)$	skewed logistic
		Shannon		
8	Section A.3	non-symm.	$\alpha_{1}u\frac{u^{\alpha }-1}{1-\alpha }+\alpha_{2}\left(1-u\right)\frac{{\left(1-u\right)}^{\alpha }-1}{1-\alpha }$	skewed Tukey λ
		Havrda-Charvát		
9	Section A.4	GMaxEnt	$u\left(u^{\alpha_{1}-1}-1\right)/\left(1-\alpha_{1}\right)$	general. Tukey λ
			$+\left(1-u\right)\left({\left(1-u\right)}^{\alpha_{2}-1}-1\right)/\left(1-\alpha_{2}\right)$	
10	Section A.5	GMaxEnt	$2\gamma^{3}/\left(1+\gamma^{2}\right)f_{N}\left(F_{N}^{-1}\left(u\right)\right)\cdot I\left(u\leq \gamma^{2}/\left(1+\gamma^{2}\right)\right)$	skewed normal
			$+2/\left(\gamma \left(1+\gamma^{2}\right)\right)\left(f_{N}(F_{N}^{-1}\left(0.5\left(\left(1+\gamma^{2}\right)u+1-\gamma^{2}\right)\right)\right.$	
			$\left.-1/\sqrt{2\pi }\right))\cdot I\left(u>\gamma^{2}/\left(1+\gamma^{2}\right)\right)$	
Fixed mean and *k*-th moment, with Q(0)=0				
11	Section A.6	Shannon	−(1−u)ln(1−u)	Weibull
12	Section A.7	Havrda-Charvát	$-\left(1-u\right)({\left(1-u\right)}^{\alpha -1}-1)/\left(\alpha -1\right)$	ext. Burr XII

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
