# Peer review of "(Generalized) Maximum Cumulative Direct, Residual, and Paired Φ Entropy Approach"

_entropy, 2020, doi:10.3390/e22010091_

Round 1

Reviewer 1 Report

Authors are dealing with several entropy notions. Results are interesting and no doubt worth being published. Strengtheness of presentation and correctness in all details, however, need to be increased. I suggest mayor revision according to the following points.

l.27    E_S(f)

l.28    instead of "under the constraint of observed" it be "under the constraint of fixed"

more general: measure theoretical or probabilistic notions have to be distinguished from their empirical or statistical counterparts throughout the paper

l.39 (and throughout the paper) type of citation does not sound good: "[60] were the first" should be replaced with "Rao et al.[60] were the first" or "authors of [60] were the first"

l.64-65 entropies are aimed by the authors to measure information what creates a confusion in terminology

l.93  Authors sentense "It is obvious to measure the amount of uncertainty of X by...(two integrals)" needs explanation: w.r.t. which quantity uncertainty is defined here ?

-does one start with some axioms

-or is it just a heuristical argument to start with the two integrals given there

-or is there another definition ?

l.188 in the last integral, dependence of F is not made visible

l.225 Start of proof of Theorem1: there are contradicting integration variables x and u

l. 266 Statement of Theorem 2 needs reformulation: It's not correct to write that (12) is a solution and only later in (13) to restrict the validity of this result because this means that saying that (12) provides the result was wrong.

l.308 there is an obvious typos in Corollary 4

l.381 there is an obvious typos in Figure 7: \lambda2

l.381 does Figure 7 or the derivations leeding to it allow additional discussion?

the densities in the upper figure look like power exponential (or p-generalized normal) densities, can you say something about this case? the densities in the lower figure look like extreme value densities, can you say something about this case?

l.407 Similarly: does Figure 8 or the derivations leeding to it allow additional discussion?

Because of the big role skewed normal distribution plays in literature: can you give explanation of the underlying entropy measure?

Author Response

Thank you very much for your very valuable comments. Please see the attached document.

Reviewer 2 Report

Review of "(Generalized) Maximum Cumulative Direct, Residual, and Paired F Entropy Approach" by Klein & Doll (2019)

The authors developed an interesting paper related to generalized maximum entropy principle (GMaxEnt) based on cumulative entropies. They obtained important results for skewed logistic, the skewed Tukey lambda and extended Burr XII distributions.

As they recall, in the literature exist only some few maximum entropy distributions derived for cumulative entropies. For this reason, I think that this manuscript has merit for publication in Entropy. However, some details must to be addressed by the authors to following the publication's process. Please, consider the following comments and references to increase the discussion of this paper:

1. The authors considered the GMaxEnt principle for the skewed distributions. However, the work of de Souza & Tsallis (1997) could be included in the discussion because, they obtained the Student distribution using the GMaxEnt principle based on Tsallis entropy.

2. As in 1, the use of other entropies could be useful for derivation of special distributions. The authors obtained some skewed distributions, as a extensions of classical distributions. However, the works of Liu (2009) and Contreras-Reyes (2015) could be included in the discussion because, Liu considered the MaxEnt principle based on Nonsymmetric entropy and, Contreras-Reyes considered the Nonsymmetric entropy for a general class of skewed distributions: skew-normal, extended skew-normal, closed skew-normal, truncated skew-normal, etc. These works could extend the Fechner apporach.

References:

de Souza, A.C., Tsallis, C. (1997). Student's t-and r-distributions: Unified derivation from an entropic variational principle. Physica A 236, 52-57.

Liu, C.S. (2009). Nonsymmetric entropy and maximum nonsymmetric entropy principle. Chaos, Solitons & Fractals 40, 2469-2474.

Contreras-Reyes, J.E. (2015). Rényi entropy and complexity measure for skew-gaussian distributions and related families. Physica A 433, 84-91.

Author Response

(The authors gave the same response as above.)

Reviewer 3 Report

The paper discusses generalized cumulative, residual and paired entropy and its maximization with respect to given constraints. From this method, the authors deduce several ME distributions connected to different types of entropies. I have several comments about the paper that should be discussed:

It is first necessary to mention that entropy was first defined for discrete variables. Differential entropy can be derived as the limit of discrete entropies, but an infinite quantity must be subtracted (for Shannon case). With this, I have two questions What is the relation between discrete and continuous cumulative entropy? How this relation changes for a general phi is there still a relation between discrete and continuous entropies? What is the reason for maximizing cumulative (direct, residual, paired) entropy? I know that several authors just used it. On the other hand, for normal entropy, there exist Shore-Johnson axioms (see e.g., 10.1103/PhysRevLett.122.120601) that tell us under which conditions is the maximum entropy procedure a consistent statistical estimation method. This should be also discussed for cumulative entropies. The reason for this is the interpretation of ME distribution for cumulative entropies which might be unclear. Of course, it can be used as a tool for defining new distributions, but then we do not know what is the role in real physical systems. Regarding generalized entropy with phi, there exist classifications of entropies with different functions phi (see e.g., 10.1209/0295-5075/93/20006) and their respective MaxEnt distributions. Can authors use such type of classification for their cases and are the distributions in the classes defined there? So of the entropies have their names (Tsallis, Kaniadakis, Hanel-Thurner, etc.) what are the corresponding ME distributions? The amount of examples and derivation of different distributions makes the paper very lengthy. I suggest moving the examples to the appendix or making a table or so because the extensive number of these examples decreases the readability of the paper. Citations should be ordered according to their appearance in the text Please make the whole manuscript shorter and more striking, it is hard to go through the whole text

Author Response

(The authors gave the same response as above.)

Round 2

Reviewer 1 Report

E_S           still needs to be written as                E_S(f)

Reviewer 2 Report

2nd review of "(Generalized) Maximum Cumulative Direct, Residual, and Paired Phi Entropy Approach" by Klein & Doll (2020)

Thanks for your reply about my last two questions. Now, it is clear for me the differences of classical MaxEnt and your approach. I have only some detail that authors must to fix. The new reference added [13] is wrong, please change it by the last one recommended:

Contreras-Reyes, J.E. (2015). Rényi entropy and complexity measure for skew-gaussian distributions and related families. Physica A: Statistical Mechanics and its Applications433, 84-91.